# The Impact of Urban Rail Transit Epidemic Prevention Measures on Passengers’ Safety Perception

**DOI:** 10.3390/ijerph20054161

**Published:** 2023-02-25

**Authors:** Pengxiang Ding, Suwei Feng, Jianning Jiang

**Affiliations:** School of Public Economics and Administration, Shanghai University of Finance and Economics, Shanghai 200433, China

**Keywords:** transportation security, prevention measures, security perception, satisfaction, structural equation model, three-factor theory

## Abstract

In 2020, COVID-19 triggered concern about the safety of public transport. To meet passengers’ expectations regarding safety, the public transport department has stepped up its pandemic prevention services. Some prevention services require passengers to follow mandatory requirements. However, whether and to what extent these requirements affect passenger satisfaction with public transportation services remains unclear. This study aims to construct an integrated framework to explore the direct and indirect relationships between four constructs (regular services quality, pandemic prevention service, psychological distance, and safety perception) and passengers’ satisfaction in the context of urban rail transit services. Based on survey data collected from 500 passengers on the Shanghai Metro, this paper examines the relationships between routine service, pandemic prevention measures, safety perceptions, and satisfaction with the service. The results from the structural equation model indicate that routine service (0.608), pandemic prevention measures (0.56), and safety perception (0.05) have positive effects on passenger satisfaction. Psychological distance negatively impacts safety perception (−0.949) and has indirect effects on passenger satisfaction. Further, in order to identify the service improvements that public transportation departments should focus on, we use the three-factor theory to identify the services that should be improved: Basic factors, such as “punctual arrival of metros”, “treatment of harmful garbage”, “increasing frequency of platform disinfection”, and “measurement of station temperature” should be treated as the first priority. As the second improvement priority, “the planning of metro stations can accommodate my travel scope” can be considered. Last, public transportation departments can enhance the exciting factor by installing “metro entrance signs” when resources are available.

## 1. Introduction

COVID-19 broke out in Wuhan, China, at the end of 2019 and spread throughout the country, affecting most parts of the world. The outbreak has also adversely affected public transportation; in most Chinese cities, buses are overcrowded, making it difficult to maintain a social distance from other passengers. Thus, public transportation creates a high risk of infection, which adds to the psychological burden of travelers concerned about contracting a virus. Concern about the infection of COVID-19 by taking public transportation has affected the passenger flow, and the occupancy rate has declined significantly. For example, in Shanghai, the metro passenger count for 2020 declined to 73% of 2019, and the metro passenger count for 2022 only occupied 62% of 2019. As well as in other parts of the world, public transportation has encountered similar difficulties. A British survey conducted in May 2020 revealed that 40% of respondents would not use public transportation if they felt unsafe [1]. To meet passengers’ expectations regarding safety, the public transport department has stepped up its pandemic prevention services, including disinfection and sterilization, detection of body temperature, sanitation, and cleaning. Some pandemic prevention services require passengers to follow mandatory requirements, such as wearing masks and maintaining social distancing. However, it remains unclear whether and to what extent these requirements affect passenger satisfaction with public transportation services and mode choice [2]. Due to the continuing spread of the pandemic, it is urgent to examine people’s perceptions of safety when taking public transport, as well as their expectations and acceptance of public transport pandemic prevention services. Moreover, we need to investigate the relationship between pandemic prevention services and passengers’ perceptions of safety and satisfaction to assist public transport departments in improving service quality under budgetary constraints.

Evaluation of the quality of public transportation is critical to attracting and retaining passengers, setting strategic objectives, and making financial decisions [3]. Groups such as the low-income rely heavily on public transportation for their travel needs, so the basic service quality of public transportation must be maintained [4]. The fare, accessibility, punctuality, compartment space, and information acquisition all play significant roles in the quality of service [5,6,7]. Taking the bus involves contact with other passengers. The compartment environment may become a breeding ground for germs and a transmission site, making people feel unsafe [8]. The COVID-19 pandemic has further exacerbated this feeling of insecurity. A safe riding environment can enhance the sense of security of passengers. Thus, public transport departments must perform an effective pandemic prevention program to ensure passenger safety and have therefore increased its pandemic prevention service, standardized the conditions for taking public transportation (such as providing negative nucleic acid results and health codes in China), and required passengers to wear masks, avoid talking loudly, maintain a safe distance, and wash their hands before and after riding a bus. The impact of these pandemic prevention services and requirements on people’s perceptions of safety, riding experiences, and acceptance, however, is unknown, so a new relationship should be established to examine its mechanisms of action [9].

Furthermore, the public transportation sector’s ultimate goal is to improve its services within a limited budget. Using the traditional importance–performance analysis (IPA), it is possible to classify public transportation service elements and determine which areas need improvement [10]. Using the IPA as an underlying framework, the three-factor theory can further illuminate the asymmetric relationship between service attributes and passenger satisfaction and more accurately classify service elements [11]. For example, the basic factors of low performance and the important performance factors are the most priority options for improving public transport services [12]. During the COVID-19 pandemic, the relationship between passengers’ perception of pandemic information, psychological distance, sense of security, and satisfaction has been studied [13]. However, more attention still needs to be paid to which services should be prioritized. Since the public transport department has made a great deal of effort to increase pandemic prevention measures and improve service quality, it is particularly important to investigate how to improve passengers’ perception of safety and satisfaction in regular services and pandemic prevention services, as well as identifying the direction in which improvements can be made.

This paper aims to provide a comprehensive analysis of urban rail transit services in Shanghai during the pandemic. We divided rail transit services into regular service and pandemic prevention service and measured variables, such as passenger safety perception and psychological distance, via questionnaire surveys. Furthermore, a structural equation model is constructed to explore the relationship between satisfaction with rail transit service quality and safety perception. Lastly, we use the three-factor theory to identify the service quality attributes that affect passenger satisfaction and categorize each element, indicating the direction of urban rail transit that needs to be improved under normalized pandemic conditions. This paper is arranged as follows: Section 1 is a literature review, Section 2 is a basic assumption and model, Section 3 is implementing the questionnaire survey and data, Section 4 is empirical analysis, Section 5 is an analysis of key improvement directions of the metro, and Section 6 is a conclusion.

## 2. Literature Review

### 2.1. The Impact of COVID-19 on Passengers’ Safety Perception and Travel Patterns

Due to the COVID-19 pandemic, passengers’ perception of safety declined, resulting in uncertainty in demand for public transportation. Since the virus spread, people have become increasingly concerned about the risk of infection when traveling [14]. The passengers in public transportation are exposed to a greater number of people, increasing the risk of infection and resulting in an increased sense of insecurity [15].

At the beginning of the pandemic, the infection risk in public transportation caused people to prefer car travel to public transport. This resulted in a sharp decline in the passenger volume of public transportation. There was an even more significant increase in cycling among commuters, while walking increased most among non-commuters [16]. In Switzerland, for example, in mid-August 2020, the average daily distance traveled by residents has reduced by sixty percent, the passenger volume of public transportation has decreased by more than ninety percent, and the proportion of people who cycle has increased considerably [17]. Public transport ridership is expected to recover gradually in the post-pandemic period but will not return to its pre-pandemic level. After the pandemic, it is anticipated that the number of riders on public transport will decrease by about 20% [18]. In Scotland, more than 30% of respondents indicated that they would reduce their bus or train travel after the pandemic, and 25% indicated that they would increase the use of private automobiles [19]. China launched health emergency responses which were classified into four levels corresponding to the severity of the pandemic. In Beijing and Shanghai, metro ridership was negatively affected by the pandemic, and health emergency responses are negatively related to metro passenger volume. Only weekend metro passenger volumes were significantly affected by the number of imported cases, which refer to individuals entering a country and being confirmed as COVID-19 patients [20].

The dependence on public transport by different groups varies. College students can use bicycles, electric scooters, or on foot as alternatives to public transportation [21]. To reduce the risk of infection, the elderly and women with children tend to choose the modes that take the least time to travel [16]. It should be noted, however, that low-income groups could not choose the mode of transportation as expected, which meant they had to rely on public transportation as they did before the pandemic, even though the risk of infection was high [22].

### 2.2. Pandemic Prevention Measures of Public Transportation under COVID-19

Metros and stations must provide effective pandemic prevention services, such as maintaining social distancing, wearing masks, and maintaining the cleanliness of vehicles during the COVID-19 pandemic [23]. Firstly, keeping social distance in an environment with severe community transmission can effectively reduce direct contact, especially on the metro, where it is possible to reduce the risk of infection by maintaining social distance [24]. Secondly, masks will also affect people’s choice of public transportation, regardless of whether they are mandatory. A study of London metro passengers found that 100% vaccination and mandatory mask-wearing had an equal impact on improving people’s preference for public transportation. London metro passenger traffic has recovered primarily due to the pandemic prevention requirement of mandatory mask-wearing. Therefore, wearing masks should be a government mandate [25]. There is a higher acceptance of mandatory measures (such as mask requirements and 1.5 m distance regulations) among the elderly and highly educated. These enforcement measures are considered helpful in improving public transportation safety. A mask’s positive effect decreases with increasing travel distance, perhaps due to the discomfort caused by the mask itself [25]. Finally, cleaning public transport vehicles and stations is also an important component of mitigating risk perceptions [26,27]. Disinfectants should be used at stations and vehicles, and ultraviolet equipment can be installed at stations [24].

### 2.3. The Impact of COVID-19 on Public Transport Satisfaction

In the long run, passenger satisfaction will determine their mode choice; Therefore, improving services that influence passenger satisfaction will increase transit occupancy rate [27]. These services relate to reduced travel costs, the cleanliness of vehicles, departure frequency, punctuality, accessibility, congestion level of streets, the safety of bus stops, noise, air quality, temperature, and crime rate [28]. Further, the improved integration of other modes with public transportation can enhance passenger satisfaction on an urban transportation network [21]. The factors affecting passenger satisfaction have changed significantly since the outbreak of the COVID-19 pandemic. It was found, for example, that the attention to the pandemic information with a closer psychological distance aggravated the anxiety of passengers, resulting in decreased feelings of safety and satisfaction during travel; in contrast, safe, reliable, and easy-to-access information about travel can enhance passenger satisfaction [13,29]. As a result of the pandemic, passengers will perceive more congestion, resulting in a reduction in satisfaction, and the value of seats will increase significantly [30].

### 2.4. The Model for Satisfaction Evaluation of Public Transportation

For public transportation operators, passenger satisfaction can provide valuable information regarding service quality. By identifying the attributes that affect passenger satisfaction, operating agencies can develop targeted strategies to improve their services [12]. The direction of service improvement is particularly crucial when the pandemic affects transit ridership. Many studies have used structural equation modeling to examine the factors influencing satisfaction with public transportation [31,32,33,34]. For example, ref. [31] adopted structural equation modeling to verify the causal relationship between Suzhou Rail Transit passenger satisfaction and its influencing factors. Ref. [29], based on the same model, showed that service qualities such as comfort, safety, reliability, and access to information helped increase loyalty to transit systems during COVID-19. Some studies also combined structural equation modeling with other econometric models to analyze public transportation satisfaction. Based on the interview data, [14] used a structural equation model and Logistic regression to analyze the satisfaction of low-income groups in using public transportation systems and found that risk concerns and risk perception had a greater impact on their satisfaction.

Derived from Frederick Herzberg’s motivator–hygiene theory [33], the three-factor theory is also widely used in the study of public service satisfaction. Based on the theory, service quality attributes can be separated into three categories: (Figure 1): basic factors, performance factors, and exciting factors [34].

Basic factors: the relationship between the basic factors and overall customer satisfaction is asymmetric. When performance is poor, it will reduce customer satisfaction, while good performance will not increase it.Performance factors: in contrast to basic factors, the relationship between performance factors and customer satisfaction is linearly symmetrical, which has a significant impact on both good performance and poor performance.Exciting factors: those attributes are asymmetrically related to customer satisfaction, which can have a significant impact if the attributes are satisfactory; that is, good performance will increase customer satisfaction, whereas failure to meet expectations will not adversely affect customer satisfaction.

Ref. [35] first applied the model to the study of transportation service satisfaction, demonstrating a nonlinear relationship between the level of satisfaction with BRT service and its quality. Based on the three-factor theory, [36] identified attributes that determine the quality of service for buses, BRTs, and vans. By utilizing the same method [11] found that Guangzhou’s public transportation system should be evaluated on the basis of service satisfaction to provide better bus service and BRT services. Due to its ability to identify type differences and structural changes in samples, dummy variables are often used to evaluate the impact of different attribute variables on satisfaction. Using the dummy variable method [12] found that the improvements focus on conventional public transportation and BRT services, as well as examined the asymmetric impact of performance factors on satisfaction with BRT services. The dummy variable method is employed in this paper to identify the improvement direction of metro routine service and pandemic prevention service.

Through the literature review, we identified three gaps that motivated this research: (1) previous literature evaluating transit services has focused only on the relationship between routine services and passenger perceived satisfaction. When there is an unexpected public health crisis, the transportation sector introduces new pandemic prevention services. The relationship between these pandemic prevention services and passenger satisfaction has not been explored. (2) Previous literature has only explored the pandemic prevention measures needed to improve transit services during a pandemic based on qualitative analysis, but there is a lack of quantitative studies exploring the effectiveness and necessity of these pandemic prevention measures to reduce people’s safety concerns. (3) Methodologically, previous studies have used structural equation modelling to verify the interactions among service factors but have been unable to suggest more specific directions for service improvement. Other studies used a three-factor theory analysis of the elements of service improvement, but the validity of the theoretical analysis framework could not be verified. In conclusion, few studies have combined the two approaches to conduct a holistic study of the relationships between service factors and directions for improvement.

To fill the knowledge gap, this paper introduces pandemic prevention services into the traditional metro service evaluation model and constructs a new evaluation model under a public health crisis. Then, based on the Shanghai Metro passenger survey data, we use structural equation modeling to explore the mechanism of improving basic and pandemic prevention services in the public transportation sector to reduce people’s concern about viruses during a public health crisis. It can also verify whether this paper’s metro service evaluation model is scientifically valid. Further, we use the three-factor theory to empirically identify more specific key influencing factors to scientifically reflect the relationship between metro services and passengers’ needs, which can enable service providers to allocate resources more effectively and ultimately increase passenger satisfaction. Using structural equation modeling and three-factor theory approaches, we can analyze the macro interaction mechanism between the elements of transportation services and identify more specific ways to improve them.

## 3. Hypothesis and Model

This paper investigates whether urban metro pandemic prevention services can improve passengers’ perceptions of safety and satisfaction, using Shanghai Metro services during the pandemic as a case study to investigate possible improvement directions of rail service quality. Our first step is establishing a conceptual model, as shown in Figure 2, and then collecting data through a questionnaire survey. We then use structural equation modeling to test the conceptual model’s hypotheses and determine the path coefficients between the variables.

### 3.1. Hypothesis

The public transportation sector has added new protective measures and requirements (wearing masks, taking body temperatures, keeping a safe distance, etc.) in order to ensure passengers are not at risk of pandemics. However, the impact of these pandemic prevention measures on passengers’ perception of safety and satisfaction is still unclear. We have categorized rail transit services into regular and pandemic prevention services in this paper investigated how these two types of services affect passenger safety perceptions and satisfaction and identified the contribution each service makes to improving passenger safety perceptions and riding experiences. Accordingly, this paper assumes that the better the regular and pandemic prevention services are, the better people’s perception of safety will be. The following assumptions are made:

**Hypothesis 1:** 
*Routine service is positively correlated with passenger perceptions of safety.*


**Hypothesis 2:** *Pandemic prevention service is positively correlated with passenger perception of safety*.

In addition, objective factors, such as population, environment, and socio-economics, and psychological factors, also influence people’s perceptions of safety [37]. Psychological distance is the subjective perception of how far something is from oneself [38]. In the context of the COVID-19 pandemic, psychological distance refers to the degree to which people fear infection when using public transportation. People may feel that the closer they are to the pandemic, the more nervous they will be and the more concerned they will be about taking public transportation, which eventually impacts their travel choices. Therefore, we have the following hypothesis:

**Hypothesis 3:** *Passengers’ psychological distance is negatively correlated with their perception of safety*.

Research on satisfaction with public transport services is quite well established [39], yet the mechanisms of how pandemics affect people’s perceptions of safety and satisfaction remain to be explored. Passenger satisfaction is defined as the perception and evaluation of transport service performance and is mainly used to measure the quality of public transport from the passenger’s perspective [40]. The quality of service is generally considered to be a prerequisite to customer satisfaction. Service quality is a subjective cognitive judgment that includes thinking and judging, while satisfaction is an affective judgment that includes liking and pleasure. Therefore, the cognitive understanding of service quality will determine the emotional judgment of satisfaction a priori. Regular service quality, such as punctuality and station cleanliness, provides passengers with a positive riding experience, which in turn improves their subjective evaluation of the entire trip [41]. In Maslow’s theory of needs, security is regarded as the second level of needs and is generally considered positively correlated with overall satisfaction [42]. Passengers will give higher satisfaction ratings if they perceive the mode of transportation to be safe [43]. Therefore, this paper has the following assumptions.

**Hypothesis 4:** *Passengers’ perception of safety is positively correlated with their satisfaction*.

**Hypothesis 5:** *Regular service quality is positively correlated with passenger satisfaction*.

**Hypothesis 6:** *Pandemic prevention service is positively correlated with passenger satisfaction*.

Figure 2 shows the theoretical model of the present research.

### 3.2. Model

This paper used structural equation modeling to examine the relationship between metro service quality and passenger satisfaction [6]. A structural equation model is a multivariate statistical method that combines factor analysis and path analysis. This method identifies the direct or indirect effects of independent variables on dependent variables by examining the relationship between latent variables, observable variables, and error. It estimates the parameters using the method of maximum likelihood estimate. A structural equation model consists of two basic models, a structural model and a measurement model. In the measurement model, there are latent and observable variables, and latent variables are abstract concepts constituted by observable variables and reflected by information obtained from observable variables [44].

In this paper, with reference to the national standard document “Evaluation Method of Urban Public Transportation Passenger Satisfaction” (GB/T 36953.1-2018), we chose 11 indicators such as fare, transit operation, and riding environment to measure the regular services. In addition, through our field research and expert interviews, this paper used nine indicators, such as wearing masks, taking body temperature, and seat spacing, to measure the prevention services. The latent variables of the structural equation need to be measured by multiple observable variables, as shown in Table 1. Referring to the study of [13], psychological distance and perception of safety were measured by three variables in Table 1. Passenger satisfaction with the metro was measured by “overall satisfaction” and “recommending your family and friends to take the Shanghai Metro”. The questionnaire was based on a 5-point Likert scale (1–5, from very dissatisfied to very satisfied).

## 4. Data

### 4.1. Participants

According to the Shanghai Comprehensive Transportation Operation’s Annual Report for 2020, 58% of Shanghai residents commute by metro, and the respondents used in this study are representative. We selected passengers from 15 lines of the Shanghai Metro system as survey respondents and contracted Dataway Inc. (8/F, 878 East, Building 1, No. 24 Jiuxianqiao Middle Road, Chaoyang District, Beijing, China) to collect samples using an online questionnaire. The survey was conducted from 21 January 2021 to 24 January 2021 and lasted for 4 days. The questionnaire consisted of three parts. The first part consisted of demographic characteristics such as age, gender, income, and education level. The second part consisted of the 28 question items in Table 1, and the third part included variables associated with travel behavior, such as frequency, duration, and purpose. The actual level of passenger flow of Shanghai Metro lines was matched with a sample collected from each line based on the average daily passenger flow for the Shanghai Metro in December 2020 released by Shanghai Transportation Commission, ensuring that the sample was representative. Finally, 500 valid questionnaires were obtained.

According to the survey, 54.0% of the sample was male, and 46.0% was female. A total of 26.0% of passengers were between the ages of 21–30, followed by 25.0% and 24.4% of those between the ages of 31–40 and 41–50, respectively. In accordance with the actual population of metro riders, the age distribution of the sample is between 11 and 50 years of age. Based on the passenger flow, passenger gender, and passenger age of each line, the sample data are of high quality and generally in line with reality.

### 4.2. Reliability and Validity Test

In this paper, we used structural equation modeling to test the relationship between routine services, pandemic prevention services, perceived safety, and satisfaction. The first step is to identify whether the observed variables are valid in the latent variable by calculating the Cronbach coefficient. We then conducted an exploratory factor analysis to determine the model’s fitness. Finally, structural equation models were analyzed using SPSS 25.0 along with AMOS 24.0 software.

(1) Reliability test. In order to verify the validity of each observable variable in the latent variable, we calculate the Cronbach coefficient. The coefficient of 0.9 indicates a very credible question. A coefficient between 0.7 and 0.9 indicates a relatively credible question. A coefficient between 0.5 and 0.7 indicates a generally credible question. A coefficient below 0.5 indicates that the questions need to be readjusted. The overall Cronbach coefficient for the scale was 0.702, and the scale reliability was satisfactory. In Table 2, the Cronbach coefficients for each latent variable passed the test, and the variables were consistent.

(2) Validity test. Validity tests were conducted using KMO and Bartlett’s spherical test for significance probability. Correlation is considered high if the KMO value is greater than 0.7, the probability of significance of Bartlett’s spherical test is greater than 0.001, and the cumulative contribution of variance is greater than 40%. Table 2 indicates that latent variables can effectively explain the observable variables.

## 5. Results

### 5.1. Structural Equation Model

As shown in Table 3, we used the data to test the model fit, and the criteria for the fit and the results are presented. The absolute fit indices and incremental fit measures met the criteria, and the selected model successfully explains the relationships between the constructs.

The results are shown in Table 4 and Figure 3. First, we found that both routine services and pandemic prevention services had a positive effect on satisfaction, supporting hypotheses H1 and H2. In addition, it is important to note that although routine services remain the most important influence on satisfaction (*β* = 0.608, *p* < 0.001), pandemic prevention service evaluation also plays a significant role in determining satisfaction (*β* = 0.56, *p* < 0.001). Secondly, perceived safety and satisfaction are also positively correlated, thus supporting hypothesis H3. It is important for the public transportation sector to focus on improving satisfaction in the post-pandemic period by providing good pandemic prevention services as well as regular services. Then, both regular and pandemic prevention services positively impact the perceived safety, and the impact of pandemic prevention services is relatively larger (*β* = 1.003, *p* < 0.001), and hypotheses H4 and H5 are supported. Last but not least, hypothesis H6 was also supported. Psychological distance had a negative effect on the perception of safety (*β* = −0.949, *p* < 0.001), and the greater the psychological distance, the less safe people feel.

### 5.2. Improvement Priorities for Services

In the structural equation model above, it was verified that regular and preventative services have a positive effect on satisfaction. This section uses the three-factor theory to identify Shanghai Metro service improvement priorities to assist the transit sector in improving satisfaction.

We can identify the asymmetric effects of different factors on satisfaction through the three-factor theory, which decomposes service attributes into three factors [36]: (1) Basic factors: These attributes reduce customer satisfaction when they perform poorly, but they do not increase customer satisfaction when they perform well. (2) Performance factors: These attributes have a significant impact on customer satisfaction based on their performance, which is linearly and symmetrically related to customer satisfaction. (3) Exciting factors: When these attributes are performed well, customer satisfaction increases. When they perform poorly, however, they do not adversely affect overall satisfaction. These factors have a non-linear relationship with satisfaction. This paper uses the three-factor theory in the following manner.

First, we recoded the metro service into two dummy variables that denote “high performance” and “low performance”. In Table 5, we have coded “very satisfied” as 1 for the high-performance dummy variable, while “average”, “unsatisfied”, and “very dissatisfied” are coded as 1. Based on the fact that most respondents considered the metro service to be good and few indicated that they were dissatisfied or very dissatisfied, we refer to [12] used “satisfied” as the reference category.

Second, to identify the classification of each service attribute and its ranking on its performance. This study regressed 40 service element dummy variables on overall satisfaction using the Ordered Logit model, and the regression results are presented in Table 6.

In the third step, we classify the service attributes based on the regression results. These variables are classified as follows: A factor can be categorized as a basic factor if a low-performance dummy variable is significant, whereas a high-performance dummy variable is insignificant. When the low-performance dummy variable is not significant, and the high-performance dummy variable is significant, this factor can be classified as an exciting factor. An attribute belongs to the performance factor if both dummy variables have significant coefficients. An attribute belongs to the insignificant factor if both of its dummy variables are insignificant.

According to the regression results and classification rules, the classification of the three factors is shown in Table 7: (1) Basic factors: Personnel health register, Metro station facilities, timely isolation of suspected persons, I can find the schedule of the metro I want to take, station disinfection, arrival on time, hazardous waste disposal, temperature measurement. Of these, five are pandemic prevention services, and three are routine services. (2) The performance factor: The planning of metro stations can cover my travel area, is a regular service. (3) The exciting factor: Clear entry signs, arrival guidance, and seat spacing are regular services.

Finally, by classifying each service attribute and average performance, this paper can depict a metro service improvement priorities hierarchy. According to the three-factor theory, the poorly performing basic factors have the greatest negative impact on passenger satisfaction and should therefore be prioritized. As shown in Table 8, the study found that “station disinfection”, “arrival on time”, “hazardous waste disposal”, and “temperature measurement” as the top priority for improvement. As long as the performance factors are improved, passenger satisfaction will increase; The performance factors are defined as the second priority, i.e., “the planning of metro stations can cover my travel range” as the second priority for improvement. If resources are available, exciting factors such as “clear entry signs” can be improved to make riding the metro convenient, easy, and enjoyable.

## 6. Discussion

Using survey data from 500 passengers on the Shanghai Metro, we find that both routine services and pandemic prevention services can improve passengers’ satisfaction and safety perception during the pandemic.

First, consistent with previous literature [2,11,13], we find that routine and pandemic prevention services can improve passenger satisfaction. Routine service remains the primary determinant of satisfaction during a pandemic, and passengers did not seem to oppose some mandatory pandemic prevention measures. Nevertheless, some studies have found that the COVID-19 countermeasures adopted by the public transport sector negatively affect travelers’ travel decisions [16]. In this regard, it is imperative that public transportation departments strike a balance between conventional services and pandemic prevention. Additionally, there is a positive correlation between passenger satisfaction and perceptions of safety. People are more likely to be satisfied with metro services when they feel safe. This finding is similar to this study [13]. Using a survey of passengers in eight Chinese cities during the pandemic, they also found a positive relationship between passengers’ perceptions of safety and satisfaction with public transportation.

Second, providing high-quality services can improve the passengers’ perception of safety. Routine and pandemic prevention services are included in these services. This finding suggests that the public transport department can enhance passengers’ perception of safety and attract passenger flows by providing effective pandemic prevention services. Furthermore, following the previous literature [13,45], this paper uses the concept of psychological distance to measure passengers’ subjective perceptions of the proximity of the pandemic. We find that passengers are more concerned about their safety when they believe they are closer to the pandemic. Based on these findings, it appears that the public transportation sector can mitigate this concern to some extent through the provision of high-quality services, especially in the area of pandemic prevention.

Finally, using the three-factor theory, we conclude that improving public transportation services should be prioritized as follows: As a first priority, it is imperative to increase the frequency of platform disinfection, the punctuality of metro arrivals, the disposal of hazardous waste, and the measurement of the temperature at the entrances. The second priority is to ensure that metro stations are planned in a manner that can accommodate the travel range of passengers. The finding implies that metro planning is also important in influencing passengers’ choice to ride the metro, and they expect more lines to be covered. This may be due to the rapid urbanization and expansion of urban land in China in recent years. However, the construction of metro stations is lagging behind, not meeting the transportation needs of the population adequately [46]. To accommodate continuous urban expansion, local governments should plan more metro lines. At the same time, the optimal construction plan should be sought in the planning. For example, when planning to build subways, coordination with existing bus systems is needed to enhance connectivity [45,47]. Furthermore, a transit-oriented concept can be adopted to promote the harmony of land use and route planning in order to provide service to more people. Last but not least, we need to improve the exciting factor of the entrance sign. The suggestions listed above will help rail transit departments improve passenger satisfaction.

This study has some limitations. First, this paper does not include potential passengers. For instance, many travelers typically have two modes of transportation (public transportation or a car). Consequently, they will also have authentic experiences and opinions regarding the quality and satisfaction of public transportation services. Future research could focus on the travel attitudes of these potential passengers, which may lead to more accurate improvement of public transportation services. Second, the cross-sectional data used in this study cannot track the effects of changes in regular and pandemic prevention services on people’s perceptions of safety and satisfaction. For example, metro pandemic prevention measures are constantly changing as the pandemic progresses. Future research should investigate whether different intensity levels may affect passengers’ perceptions of safety and satisfaction with the service. Finally, this paper only examines Shanghai as a case study, and future research could collect data from a number of cities for comparative analysis.

## 7. Conclusions

The present study investigated the relationships among regular services, prevention services, psychological distance, safety perceptions, and satisfaction with public transportation. The findings indicate that both metro’s regular and preventive service quality positively affect overall satisfaction, with regular service quality having a greater effect. Additionally, we also found that both routine service and pandemic prevention measures are positively associated with passengers’ perceptions of safety. People’s perception of safety in the metro is also affected by psychological distance; the enhancement of psychological distance will decrease people’s perception of safety in the metro. Lastly, passengers’ perception of safety is positively related to satisfaction.

Accordingly, some implications are provided for the transportation sector: during a public health crisis, passengers’ perceptions of safety can significantly impact their riding experience and satisfaction. This emphasizes the need for public transportation providers to prioritize safety as a key factor in improving overall passenger satisfaction. Furthermore, rail transit’s regular services need to be continuously improved. During a pandemic, passengers are concerned about both safety and service quality. Specifically, utilizing the three-factor theory, this study suggests that the transit sector must improve the quality of pandemic prevention services by taking temperature measurements at entry stations, increasing disinfection frequency, and disposing of hazardous waste. Moreover, to enhance the convenience of the public transportation network and improve the quality of regular services, pictorial guidance and signage should be installed near subway stations. Nevertheless, it should be noted that our analysis of the improvement priorities for metro services based on the three-factor theory assumes that the costs associated with improving each service remain unchanged. In practice, if two services have similar improvements in satisfaction, the metro company should prioritize the low-cost elements first.

## Figures and Tables

**Figure 1 ijerph-20-04161-f001:**
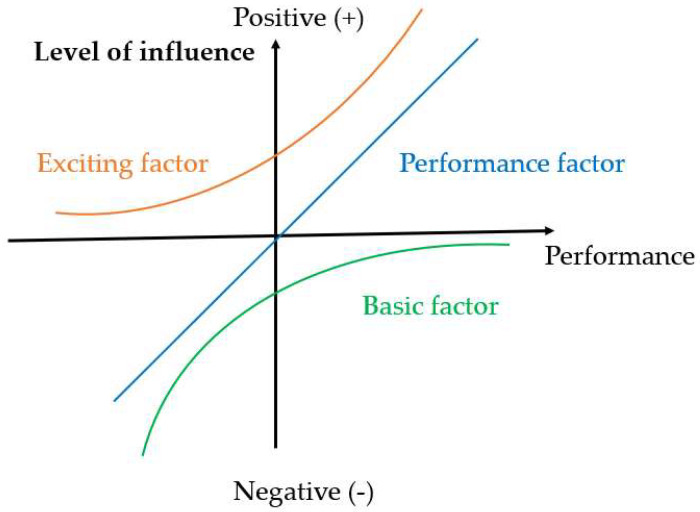
Three-factor theory (adapted from Kano et al., 1984) [34].

**Figure 2 ijerph-20-04161-f002:**
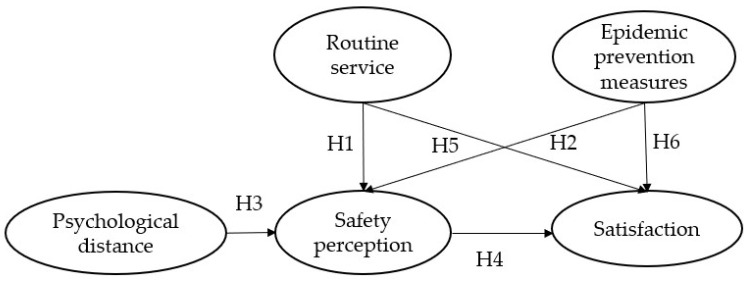
Conceptual model and assumptions. Note: H1–H6, Hypothesis 1–Hypothesis 6.

**Figure 3 ijerph-20-04161-f003:**
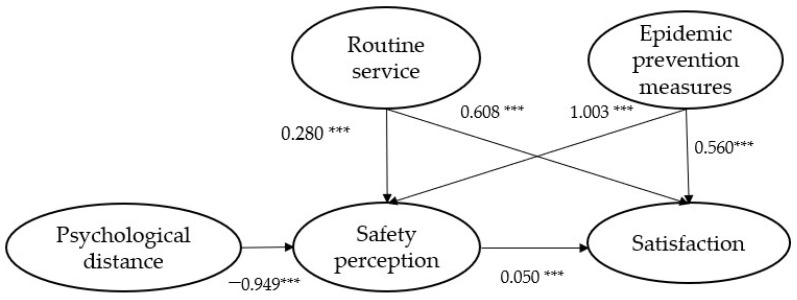
Structural equation model. Note: *** is significant at *p* < 0.001 level.

**Table 1 ijerph-20-04161-t001:** Latent and observable variables.

Latent Variables	Observable Variables
Regular services	Fare affordability
	Metro station facilities
	The carriage is clean, bright, and tidy
	The planning of metro stations can cover my travel area
	I can find the schedule of the metro I want to take
	Ride safely and orderly
	Safety guaranteed
	Arrival on time
	The staff is friendly and welcoming
	Clear entry signs
	Arrival guidance
Pandemic prevention services	Seat spacing
	Wearing a mask throughout
	Personnel health register
	Cabin sterilization
	Timely isolation of suspected persons
	Hazardous waste disposal
	Staff guidance
	Temperature measurement
	Station disinfection
Psychological distance	COVID-19 is easily spread in the metro
	COVID-19 carriers may be in the metro car
	COVID-19 will be a long-term threat to metro travel
Safety perception	I do not worry about getting infected when I am waiting at a metro station
	I do not worry about contagion in the metro
	I do not worry about getting infected when I ride the metro
Satisfaction	How satisfied are you with the Shanghai Metro in general?
	Would you recommend your family and friends take the Shanghai Metro?

**Table 2 ijerph-20-04161-t002:** Reliability and validity test.

Latent Variables	Cronbach’s Alpha	KMO	Bartlett	Cumulative Variance
Regular Services	0.702	0.812	*p* < 0.001	43.23%
Pandemic Prevention Services	0.614	0.704	*p* < 0.001	51.42%
Psychological Distance	0.915	0.761	*p* < 0.001	85.64%
Safety Perception	0.933	0.765	*p* < 0.001	88.19%
Satisfaction	0.701	0.500	*p* < 0.001	77.74%

Note: KMO, Kaiser Meyer Olkin.

**Table 3 ijerph-20-04161-t003:** Model fit.

Fit Indices	Absolute Fit Indices		Incremental Fit Measures	
Index	CMIN/DF	RMSEA	CFI	IFI
Structure model	2.219	0.049	0.908	0.909
Recommended value	<3	<0.5	>0.9	>0.9

Note: CMIN/DF, Chi-square to degrees of freedom ratio; RMSEA, mean square error of approximation; CFI, comparative fit index; IFI, incremental fit index.

**Table 4 ijerph-20-04161-t004:** Path coefficient and hypothesis verification results.

Path	Coefficient (*β*)	S.E.	*p*	Hypothesis
Satisfaction ← egular Services	0.608	0.085	***	H1
Satisfaction ← Pandemic Prevention Services	0.56	0.16	***	H2
Satisfaction ← Safety Perception	0.05	0.018	***	H3
Safety Perception ← Regular Services	0.28	0.095	***	H4
Safety Perception ←Pandemic Prevention Services	1.003	0.268	***	H5
Safety Perception ←Psychological Distance	−0.949	0.041	***	H6

Note: *** is significant at *p* < 0.001 level S.E., estimate of the standard error of the covariance.

**Table 5 ijerph-20-04161-t005:** Coding of independent variables.

Categories	High Performance Dummy Variable	Low Performance Dummy Variable
Excellent	1	0
Good	0	0
Fair	0	1
Poor	0	1
Unacceptable	0	1

**Table 6 ijerph-20-04161-t006:** Dummy variables regression.

Variables	Coef.	St.Err.	*t*-Value	*p*-Value	Sig.
Fare affordability (High)	−0.165	0.255	−0.65	0.517	
Fare affordability (Low)	0.459	0.309	1.49	0.137	
Metro station facilities (High)	0.022	0.273	0.08	0.936	
Metro station facilities (Low)	−0.539	0.281	−1.92	0.055	*
The carriage is clean, bright, and tidy (High)	0.421	0.276	1.53	0.127	
The carriage is clean, bright, and tidy (Low)	−0.146	0.283	−0.52	0.606	
The planning of metro stations can cover my travel area (High)	−0.609	0.276	−2.21	0.027	**
The planning of metro stations can cover my travel area (Low)	0.487	0.292	1.67	0.095	*
I can find the schedule of the metro I want to take (High)	0.382	0.276	1.39	0.165	
I can find the schedule of the metro I want to take (Low)	−0.561	0.258	−2.17	0.03	**
Ride safely and orderly (High)	0.433	0.318	1.36	0.173	
Ride safely and orderly (Low)	−0.083	0.277	−0.3	0.763	
Safety guaranteed (High)	0.38	0.304	1.25	0.211	
Safety guaranteed (Low)	−0.347	0.275	−1.26	0.208	
Arrival on time (High)	−0.226	0.27	−0.84	0.402	
Arrival on time (Low)	0.615	0.32	1.92	0.054	*
The staff is friendly and welcoming (High)	−0.404	0.287	−1.41	0.159	
The staff is friendly and welcoming (Low)	−0.417	0.261	−1.6	0.109	
Clear entry signs (High)	0.588	0.316	1.86	0.063	*
Clear entry signs (Low)	−0.266	0.286	−0.93	0.354	
Arrival guidance (High)	0.583	0.285	2.04	0.041	**
Arrival guidance (Low)	0.04	0.27	0.15	0.882	
Seat spacing (High)	−0.585	0.271	−2.16	0.031	**
Seat spacing (Low)	0.068	0.338	0.2	0.84	
Wearing a mask throughout (High)	−0.106	0.292	−0.36	0.716	
Wearing a mask throughout (Low)	0.065	0.306	0.21	0.833	
Personnel health register (High)	−0.137	0.286	−0.48	0.633	
Personnel health register (Low)	−1.255	0.293	−4.28	0	***
Cabin sterilization (High)	−0.179	0.297	−0.6	0.548	
Cabin sterilization (Low)	−0.315	0.269	−1.17	0.24	
Timely isolation of suspected persons (High)	−0.202	0.302	−0.67	0.503	
Timely isolation of suspected persons (Low)	0.529	0.299	1.77	0.078	*
Hazardous waste disposal (High)	0.04	0.324	0.12	0.902	
Hazardous waste disposal (Low)	−0.575	0.281	−2.04	0.041	**
Staff guidance (High)	0.124	0.338	0.37	0.713	
Staff guidance (Low)	−0.175	0.255	−0.69	0.491	
Temperature measurement (High)	−0.123	0.355	−0.35	0.73	
Temperature measurement (Low)	0.538	0.26	2.07	0.039	**
Station disinfection (High)	−0.393	0.272	−1.45	0.148	
Station disinfection (Low)	0.573	0.306	1.87	0.061	*
Mean dependent var	3.944				
Pseudo R-squared	0.24				

Note: *** is significant at *p* < 0.001 level; ** is significant at *p* < 0.01 level; * is significant at *p* < 0.05.

**Table 7 ijerph-20-04161-t007:** Classification and ranking of service elements.

Service Attributes	Factor	Mean	Rank
Personnel health register	Basic	4.000	3
Metro station facilities	Basic	3.990	4
Timely isolation of suspected persons	Basic	3.970	6
I can find the schedule of the metro I want to take	Basic	3.870	9
Station disinfection	Basic	3.840	13
Arrival on time	Basic	3.830	14
Hazardous waste disposal	Basic	3.660	18
Temperature measurement	Basic	3.440	20
The planning of metro stations can cover my travel area	Performance	3.860	11
Clear entry signs	Exciting	3.910	8
Arrival guidance	Exciting	3.870	10
Seat spacing	Exciting	3.710	17

**Table 8 ijerph-20-04161-t008:** Improvement priorities based on the three-factor theory.

Service Quality Factors
**Top priorities—Basic factors with a mean performance lower than four**
Temperature measurement (E)
Hazardous waste disposal (E)
Arrival on time (R)
Station disinfection (E)
**Second priorities—Performance factors**
The planning of metro stations can cover my travel area (R)
**Third priorities —Exciting factors**
Clear entry signs (R)

Note: R = Regular services; E = Pandemic prevention services.

## Data Availability

The data used to support the findings of this study are available from the corresponding author upon request.

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
