# Peer review of "The Impact of Urban Rail Transit Epidemic Prevention Measures on Passengers’ Safety Perception"

_ijerph, 2023, doi:10.3390/ijerph20054161_

Round 1

Reviewer 1 Report

No major issues with this paper. An interesting and timely topic that is of ongoing importance. The three-factor theory used as a way to develop survey and analyze results is also useful. I think you could more clearly show this link throughout the paper. Also perhaps discussing why certain questions were categorized as basic, performance or exciting. Your overall conclusions are very generic... since anyone reading this would expect better basic service, good pandemic measures and higher safety perception to result in more positive passenger satisfaction. So that being said, there is nothing inherently wrong with the article as is... I'm just not sure what the added value is when it doesn't really say anything we don't already know.

Reviewer 2 Report

This study presents a study of people's perception of transit safety measures against the COVID-19 pandemic. In my opinion, that topic still needs more research. The study is appropriate; however, it needs some modifications. The authors should focus on their review of the statistical models used for the same problem and then justify their selection of the structural equation model and three factors theory, among others. Second, the authors should have their recommendations in bullet form more explicitly in the discussion section. Third, the authors should refer to the transit planning problem as it is still the foremost important factor that affects users' choice of public transportation, as found in Integrating underground line design with existing public transportation systems to increase transit network connectivity: Case study in Greater Cairo" and An optimal metro design for transit networks in existing square cities based on non-demand criterion.

Reviewer 3 Report

The research topic is important and timely. I have the following comments, which I hope will be helpful to the authors:

1.  The literature review is weak, the autor's should indicate what is the research gap in the current literature? The authors should make a clear statement.

2. Why fig. no. 3 is like "3D"?

3. The discussion section should be expanded. The authors should compare the results of their research with similar studies in the world
